# Characterization of Two Gonadal Genes, zar1 and wt1b, in Hermaphroditic Fish Asian Seabass (*Lates calcarifer*)

**DOI:** 10.3390/ani14030508

**Published:** 2024-02-03

**Authors:** Han Cui, Haoyu Zhu, Wenzhuo Ban, Yulin Li, Ruyi Chen, Lingli Li, Xiaoling Zhang, Kaili Chen, Hongyan Xu

**Affiliations:** 1Integrative Science Center of Germplasm Creation in Western China (CHONGQING) Science City, College of Fisheries, Southwest University, Chongqing 402460, China; cuih24@163.com (H.C.); zzzhy703@163.com (H.Z.); banwenz001@163.com (W.B.); liyulin0835@outlook.com (Y.L.); chenruyiod@gmail.com (R.C.); aplevu@163.com (L.L.); zhangxiaoling0602@163.com (X.Z.); 2Key Laboratory of Freshwater Fish Reproduction and Development, Chongqing 400715, China; 3Key Laboratory of Aquatic Sciences of Chongqing, Ministry of Education, Chongqing 400715, China

**Keywords:** hermaphroditic fish, gametogenesis, sex reversal, *wt1* gene, *zar1* gene, in situ hybridization

## Abstract

**Simple Summary:**

Gonadal development and gametogenesis are key processes for fish genetic breeding. Particularly, reproductive development-related genes such as zygote arrest-1 (Zar1) and Wilms’ tumor 1 (Wt1) play an important role in oogenesis, with the latter also involved in testicular development and gender differentiation. Here, the cDNA fragments of *Lczar1* and *Lcwt1b* were identified in Asian seabass (*Lates calcarifer*), a valuable hermaphrodite fish for studying sex reverse. Furthermore, their mRNA distributions in the cells of gonadal tissues were detected. *Lczar1* mRNA was exclusively expressed in the ovary, while *Lcwt1b* mRNA was majorly expressed in the gonads in a higher amount in the testis than in the ovary. Specifically, *Lczar1* mRNA was mainly concentrated in oogonia and oocytes at early stages in the ovary, but was undetectable in the testis. *Lcwt1b* mRNA was localized not only in gonadal somatic cells (testis and ovary), but also in female and male germ cells at the early developmental stages. Our study indicated that *Lczar1* and *Lcwt1b* possibly play roles in gonadal development. Therefore, the findings of this study will provide a basis for clarifying the mechanism of *Lczar1* and *Lcwt1b* in regulating germ cell development and the sex reverse of Asian seabass and even other hermaphroditic species.

**Abstract:**

Zygote arrest-1 (Zar1) and Wilms’ tumor 1 (Wt1) play an important role in oogenesis, with the latter also involved in testicular development and gender differentiation. Here, *Lczar1* and *Lcwt1b* were identified in Asian seabass (*Lates calcarifer*), a hermaphrodite fish, as the valuable model for studying sex differentiation. The cloned cDNA fragments of *Lczar1* were 1192 bp, encoding 336 amino acids, and contained a zinc-binding domain, while those of *Lcwt1b* cDNA were 1521 bp, encoding a peptide of 423 amino acids with a Zn finger domain belonging to Wt1b family. RT-qPCR analysis showed that *Lczar1* mRNA was exclusively expressed in the ovary, while *Lcwt1b* mRNA was majorly expressed in the gonads in a higher amount in the testis than in the ovary. In situ hybridization results showed that *Lczar1* mRNA was mainly concentrated in oogonia and oocytes at early stages in the ovary, but were undetectable in the testis. *Lcwt1b* mRNA was localized not only in gonadal somatic cells (the testis and ovary), but also in female and male germ cells in the early developmental stages, such as those of previtellogenic oocytes, spermatogonia, spermatocytes and spermatids. These results indicated that *Lczar1* and *Lcwt1b* possibly play roles in gonadal development. Therefore, the findings of this study will provide a basis for clarifying the mechanism of *Lczar1* and *Lcwt1b* in regulating germ cell development and the sex reversal of Asian seabass and even other hermaphroditic species.

## 1. Introduction

Gonadal development and gametogenesis are key processes for fish reproduction and breeding. Among them, reproductive development-related genes and auxiliary factors play essential roles in gonadal development and gametogenesis [1,2,3]. The development and differentiation of gonads are highly similar in animals, including the specification and migration of primordial germ cells, the differentiation of germ cells, and the formation of gametes [4]. Although it is reported that a series of genes are involved in the process of gonadal differentiation and development, only some of them have been identified so far, such as *nanos3*, *vasa* (*Ddx4, DEAD box polypeptide 4*), *dazl* (*deleted in azoospermia-like genes*), and *piwi* (*a P-element-induced wimpy testis*) [5,6,7]. In addition, some genes crucial for sex differentiation, including *cyp19a1a* (*cytochrome P450, family 19, subfamily A, polypeptide 1a*), *foxl2* (*forkhead-box protein L2*), *rspo1* (*R-spondin 1*), *dmrt1* (*dsxand mab-3-related transcription factor 1*), *amh* (*anti-Mullerian hormone*), and *sox9* (*SRY box 9*), usually participate in the process of sex reversal [8]. Recently, other genes like *zar1* (zygote arrest-1) and *wt1* (Wilms’ tumor 1) were also discovered to be closely related to gonadal development [9,10]. 

*Zar1* is an oocyte-specific and maternal gene that functions in the oocyte-to-embryo transition in mouse [11]. Zar1 protein has a conserved C12 structure (i.e., a zinc finger domain) that may be related to transcriptional regulation [10,12]. *Zar1* was first identified in a mouse (*Mus musculus*) [11]. Subsequently, *zar1* has been identified in more than ten species, including rats (*Rattus norvegicus)* [12], cattle (*Bos taurus*) [13], pigs (*Sus scrofa*) [14], rabbits (*Leporidae*) [15], chickens (*Gallus gallus*) [16], pufferfish (*Tetraodontidae*) [12], African Xenopus (*Xenopus laevis*) [17], zebrafish (*Danio rerio*) [18], etc. Zar1 has conserved ovarian localization [19], and it may also have some non-ovarian functions, because it is expressed in some non-ovarian organs, such as the lung, spleen, kidney and heart [13,14,15,20,21]; the *zar1* gene is also expressed in cancer cells [22,23,24]. In mice, the deletion of the *zar1* gene has no effect on growth and development, but can lead to an imbalance of the transcriptome of oocytes and the cessation of the development of zygotes in the two-cell stage, resulting in the infertility of female mice [19]. Furthermore, the loss of *zar1* expression in female zebrafish leads to early oocyte apoptosis, the masculinization of gonads, and sex reversal from female to male [18]. Meanwhile, in Ussuri catfish (*Pseudobagrus ussuriensis*), it is found that the *zar1* gene may be involved in oogenesis [25]. In *Xenopus laevis*, *zar1* is expressed throughout oogenesis, and its expression level is stable during oocyte maturation [17]. In rainbow trout, *zar1* may play an important role in oocyte development, meiotic control, and early embryonic development [26]. *Zar1* is expressed in the ovary and embryonic stage and functions in the oogenesis and embryonic development of the Spotted scat (*Scatophagus argus*) [27]. Japanese eel *zar1* began to play a role after ovarian differentiation, which may function in oogenesis [28]. These results indicate that *zar1* is majorly crucial for early oogenesis.

*Wt1* was reported in earlier studies on tumors, and was recently found to be involved in animal gonadal development [29,30,31]. Recent studies verify that *wt1* regulates gonadal development at various stages [9,32,33,34,35,36]. The Wt1 protein contains a conserved zinc finger structure in the C-terminal, which is responsible for gonadal development [29,30,31]. There is only one Wt1 homologue in tetrapod species, while there are two paralogs in teleost fish, namely Wt1a and Wt1b [37]. Especially, Wt1 (tetrapod Wt1; fish Wt1a and Wt1b) includes the variants of Wt1 (+KTS) and Wt1 (-KTS), being different in three amino acids (aa), e.g., KTS (lysine, threonine, and serine) [38,39,40]. In animals lacking the Wt1 (+KTS) variant, podocyte function was disturbed and sex reversal occurred, while +KTS variants were considered important regulators of the *Sry gene* in the sex determination pathway [30,39]. In mice, Wt1 plays a crucial role in spermatogenesis [41,42]; the deletion of the *wt1* gene led to kidney necrosis, gonadal dysplasia, and sex cell loss [29,42,43]. In medaka, the knockdown of *wt1* led to the abnormal development of the forekidney, and a great reduction in the number of primordial germ cells [44]. The decreased expression levels of *wt1a* and *wt1b* in zebrafish resulted in defects in the development of the kidney and pronephros, as well as embryonic glomerular and tubular cysts [38]. Although those studies have shown that *zar1* and *wt1* are essential for gonadal differentiation and development, their expressions profiles vary in different species [10,18,25,41,42], and their potential specific functions also need to be further addressed.

Asian seabass (*Lates calcarifer*) is a proandrous hermaphrodite and tropic species [45], responds to exogenous hormone stimulation [46], and has faster growth rate in the young stage [47]. These characteristics have made the Asian seabass a valuable model for investigating sex reversal in vertebrates. However, the mechanism of sex reversal initiation is still unclear. In Asian seabass, several genes related to sex reversal, such as *vasa*, *dmrt1*, and *cyp19a1a*, have been identified and well studied [48,49,50]. Based on the previous study, *zar1* was found to be exclusively expressed in the ovary while *wt1b* was predominantly involved in testicular development and sex differentiation; thus, investigations on *zar1* and *wt1b* will facilitate a demonstration of the molecular mechanism behind sex reversal in fish species. In this study, we aimed to characterize the cDNAs and aa sequences of *Lczar1* and *Lcwt1b* genes in Asian seabass, and then analyzed their expression patterns in gonadal cells during gametogenesis. Thus, our research may be helpful for further studies on the mechanism of gonadal development and germ cell development in sex reversal hermaphroditic fish.

## 2. Materials and Methods

### 2.1. Fish and Ethics 

Animal experiments in this study were performed under the guidelines and approval of the Institutional Animal Care and use Committee of Southwest University (No. IACUC-20210111-01). The sexually mature Asian seabass used in the experiment were collected from a fish farm in Hainan Haikou. The fish were anesthetized with 30 mg/L of eugenol (Solarbio, Beijing, China) before being sacrificed. A panel of tissue samples were dissected as follows: brain, heart, kidney, liver, intestine, spleen, testis, and ovary. These tissues were used for RNA extraction, frozen sections, and paraffin sections.

### 2.2. RNA Extraction and Reverse Transcription 

The total RNA was extracted from the tissues using Trizol (invitrogen, Singapore), and genomic DNA contamination was removed using DNase. The quantity and purity of RNA was examined using a NanoQ TM micro spectrophotometer, and the integrity of RNA was examined via 1% agarose gel electrophoresis. The ratio of 260/280 was 2.04 to 2.22, and the ratio of 260/230 was 1.94 to 2.46, indicating that the RNA was pure. We used the PrimeScript^TM^ RT reagent kit with gDNA Eraser (Takara, Dalian, China) (a reverse transcription kit) to reverse transcribe total RNA to synthesize the first strand of cDNA.

### 2.3. Cloning of cDNA Fragments

According to the predicted cDNA sequences of *Lczar1* (XM_018669829.1) and *Lcwt1b* (XM_018668041.2) in the NCBI database, specific primers pairs (*Lczar1-CDS*; *Lcwt1b-CDS*) were designed (Table 1) and synthesized by Tsingke Biotechnology Co., Ltd. PrimeSTAR^®^ Max DNA Polymerase was used for PCR. The amplification reaction conditions were 98 °C for 30 s, 60 °C for 30 s, and 72 °C for 60 s in 30 cycles, and 72 °C for 5 min. PCR products were detected via electrophoresis in 1% agarose gel; DNA was recovered with the Tiangen gum recovery kit and ligated into a pGEM-T-easy vector, and then the recombinant plasmid was transformed into Top10 chemically competent cells. After the screening of colonies, the positive ones were picked up and sequenced by the company (Tsingke Biotechnology Co., Ltd., Beijing, China).

### 2.4. Sequence and Phylogenetic Analysis

The open reading frames (ORFs) of *Lczar1* and *Lcwt1b* were predicted using ORF Finder (https://www.ncbi.nlm.nih.gov/orffinder/, accessed 20 March 2023), and the aa sequences were deduced using DNAMAN 6.0 software. Likewise, the conserved motifs of LcZar1 and LcWt1b were searched for using a website tool (https://www.ncbi.nlm.nih.gov/Structure/cdd/wrpsb.cgi, accessed 21 March 2023). The multiple alignment analysis of protein sequences was performed using Vector NTI 11.5 software. Phylogenetic trees were constructed by MEGA 7.0 software through the neighbor-joining (NJ) method, and the bootstrap value was set to 1000.

### 2.5. Expression Analysis of Lczar1 and Lcwt1b in Tissues via RT-qPCR

To determine the tissue expression profiles of *Lczar1* and *Lcwt1b*, RT-qPCR was performed with specific primers (Table 1) and *Lcβ-actin* (an internal control) primers (Table 1). RT-qPCR was run using *Bio*-*Rad* CFX96 Real-Time PCR Detection System (Bio-Rad, Hercules, CA, USA) with a program of 95 °C for 30 s, 95 °C for 5 s, and 60 °C for 34 s in 39 cycles, and 95 °C for 15 s, 65 °C for 5 s, and 95 °C for 15 s. The relative mRNA expression levels were calculated using the 2^−∆∆CT^ method, and the control was set as heart tissue. Statistical analysis was implemented via one-way ANOVA and Tukey in SPSS 20.0. All statistics were carried out using GraphPad Prism version 9.0 (GraphPad Software, San Diego, CA, USA).

### 2.6. In Situ Hybridization

In situ hybridization (ISH) of cryostat sections was performed as previously described [4] to detect the cellular distribution of *Lczar1* and *Lcwt1b* mRNA. Gonad tissues were fixed with 4% PFA overnight, dehydrated with gradient methanol in PBS, and stored at −20 °C until use. The samples were embedded in O.T.C. (Tissue-Tek, Torrance, CA, USA), and then sectioned using a Thermo Scientific HM325 microtome (Thermo Fisher Scientific, Waltham, MA, USA) with a thickness of 5 μm. Plasmids containing fragments of *Lczar1* and *Lcwt1b* were digested and linearized with restriction endonuclease (*Nco*I, *Nde*I, *Sac*II, and *Sal*I) (Takara, Dalian, China) for preparing antisense and sense RNA probes. Probes were synthesized using the DIG RNA labeling kit (Roche, Mannheim, Germany). The signals were developed with BCIP/NBT substrates on sections, and propidium iodide (PI) was used for nucleus staining. The images were taken by a Zeiss Axiovert upright microscope (EK 130x85 Axiovert 200, ZEISS Group, Oberkochen, Germany) and photographed using a Zeiss digital camera (Axiocam 506 color, ZEISS Group, Oberkochen, Germany).

## 3. Results

### 3.1. Sequence Alignment and Phylogenetic Analysis of zar1 and wt1b cDNA

A partial fragment of 1192 bp was isolated in Asian seabass via RT-PCR, and shared 99.08% identity with *zar1* (XM_018669829.1) deposited in the public database (NCBI). The cloned *zar1* cDNA fragment contains a partial 5′ UTR of 16 bp, an ORF of 1011 bp, and a partial 3′ UTR of 165 bp (Appendix A), encoding a peptide of 336 aa with a zinc-binding domain (Appendix A). Likewise, a cDNA fragment of 1521 bp was obtained via RT-PCR and shared 99.87% identity with *wt1b* (XM_018668041.2) retrieved from GenBank of NCBI. The isolated *wt1b* cDNA fragment was composed of a partial 23 bp 5’ UTR, a partial 226 bp 3’ UTR, and a 1272 bp ORF encoding a peptide of 423 aa with a Zn finger domain (Appendix A). 

Multiple alignment of the Zar1 protein sequences revealed that LcZar1 shared a high identity of 84% with *Monopterus albus*, and a moderate identity of 62% with *Danio rerio*. However, LcZar1 shared a lower identity of 47% with *Rattus norvegicus* (Figure 1). Furthermore, the results showed a highly conserved zinc-binding domain in the C-terminal among LcZar1 and its homologs (Figure 1). The Wt1 protein sequence alignment analysis revealed that LcWt1b shared a high identity of 95% with *Acanthopagrus latus* Wt1b and a moderate identity of 86% with *Oryzias latipes* Wt1b (Figure 2), showing that Wt1 protein has a conserved zinc finger structure (Figure 2). 

Phylogenetic analysis indicated that LcZar1 was clustered into a single clade with fish homologs, in contrast to other species (Figure 3A). LcWt1b was clustered with Wt1b homologs into a branch, while Wt1a homologs of fish and tetrapods were clustered into another branch (Figure 3B).

### 3.2. The mRNA Expression Profiles of Lczar1 and Lcwt1b

In order to examine the expression levels of *Lczar1* and *Lcwt1b* in different tissues of seabass, RT-qPCR (real-time quantitative PCR) was conducted. As shown in Figure 4A, *Lczar1* was significantly highly expressed in the ovary, in a concentration over 700 times higher than that in the heart and testis, but was barely detected in other tissues examined in this study (Figure 4A). However, *Lcwt1b* was significantly expressed highly in the testis, and the expression level of *Lcwt1b* in the testis was about 60 times higher than that in the ovary. Furthermore, *Lcwt1b* was also partially detected in other tissues including the heart, liver, spleen, and kidney examined in this study, but was scarcely detected in the intestine and brain (Figure 4B).

### 3.3. The Cellular Localization of Lczar1 and Lcwt1b in Gonads

In order to detect the cellular distribution of the *Lczar1* and *Lcwt1b* mRNA in the gonad tissues, chemical ISH was performed. The results showed that the signal of *Lczar1* mRNA was obviously observed in oogonia and oocytes at the primary growth stage, but scarcely detected in oocytes at the vitellogenic stage (Figure 5A,B), and there was no signal in the ovary detected via the *Lczar1* sense riboprobe (Figure 5C). In particular, the *Lczar1* mRNA signal was detected to be the strongest in oocytes at the primary growth stage, moderate in oogonia, and almost undetectable in oocytes at the vitellogenic stage (Figure 5A,B). Moreover, the cellular distribution of *Lcwt1b* mRNA in the ovary showed a different pattern from that of *Lczar1* mRNA. The *Lcwt1b* mRNA signal was detected in oogonia and oocytes at the primary growth stage and as well as in some somatic cells (Figure 5D,E,G–I), and there was no signal in the ovary detected by the sense riboprobes of *Lcwt1b* (Figure 5F). Particularly, the signals in oogonia and primary growth stage oocytes were much stronger, but almost no signal was detected in vitellogenic-stage oocytes (Figure 5D,E,G,H).

In the testis, the sense and antisense probe of *Lczar1* did not detect any signal in spermatogonia, spermatocytes, or spermatids (data not shown). However, the *Lcwt1b* mRNA signals detected by the antisense probe were mainly localized in spermatogonia, spermatocytes, spermatids, and some somatic cells, such as Sertoli cells, Leydig cells, and so on (Figure 6A,C,D), while the sense probe of *Lcwt1b* mRNA did not detect any signal in the testis (Figure 6B).

## 4. Discussion

In this study, LcZar1 protein had a conserved zinc-binding domain and LcWt1b protein had a conserved zinc finger structure, similarly to their homologs in different species, indicating that Zar1 and Wt1 are evolutionarily conserved. Additionally, sequence alignment showed that LcZar1 and LcWt1b had the highest homology with the orthologs in fish, and the highly conserved domain of Zar1 was the C12 structure, being essential for transcriptional regulation [10,11,51]. There is only one isoform, Wt1, in tetrapod vertebrates, while two are identified in teleost fish, namely Wt1a and Wt1b. Moreover, LcWt1b was clustered with the Wt1b homologs from other species, but obviously separated from the branch of fish and tetrapods’ Wt1a homologs, suggesting that LcWt1b is a member of the Wt1b family.

RT-qPCR analysis showed that both *Lczar1* and *Lcwt1b* were primarily expressed in the gonads, but exhibited different expression profiles in adult tissues of Asian seabass. *Lczar1* was expressed extremely highly in the ovary, which is consistent with previous studies [52], suggesting that *Lczar1* possibly functions majorly in the ovary of the Asian seabass. In eel and Spotted scat, it was found that *zar1* was strongly expressed in the ovary, but undetectable in other tissues [27,53]. In Japanese eel, *zar1* was also mainly expressed in the ovary [28]. In rainbow trout, *zar1* was predominantly expressed in the ovary and testis, with slight expression in other somatic tissues [26,54]. However, in frogs, *zar1* is highly expressed not only in the ovaries, but also in the lungs and muscles [12]. In mice, *zar1* is specifically expressed in the ovary [11,12,19]. In bovine tissues, *zar1* is expressed not only in the ovary, but also in the heart, muscle, and testis [13,20]. Some studies also showed that *zar1* is expressed in the hypothalamus part of the brain [14]. In pig tissues, *zar1* is expressed not only in the ovary, but also in the testis, pituitary gland, and part of the hypothalamus [14]. In New Zealand rabbits, *zar1* is expressed in the lung, heart, liver, spleen, kidney, ovary, and uterus, and has the highest expression in the lung [15]. In addition, studies have shown that methylation of *zar1* is associated with cancer [22,23,24]. These studies show that *zar1* is mainly expressed in the gonads of fish, while in higher animals, it shows different expression patterns. It is indicated that the spatial and temporary expressions of *zar1* are divergent among species and may have species-specific functions. Unlike *zar1*, in this study, *Lcwt1b* was significantly highly expressed in the testis. The expression of *Lcwt1b* in the testis was about 30–60 times higher than that in the ovary, heart, liver, spleen, and kidney, but *Lcwt1b* mRNA was scarcely detected in the tissues of the intestine and brain, implying that *Lcwt1b* plays a more important role in the testis than in the ovary. As a male precocious hermaphrodite fish, it is suggested that in Asian seabass, *Lcwt1b* may play a role in the process of sex reversal. *Wt1* was found to be related to tumors in earlier studies, but recent studies verify that it also functions in the development of animal gonads. In mice, *wt1* is strongly expressed in the ovary and testis, but also in the heart, spleen, kidney, muscle, eyes, and gills [37]. In zebrafish, *wt1a* and *wt1b* are strongly expressed in the ovary, testis, spleen, and kidney, and also in the heart, skin, and muscle; in addition, *wt1a* is weakly expressed in the liver, and *wt1b* is weakly expressed in eyes and gills [37]. In medaka adult tissues, Wt1a and Wt1b showed an overlapping expression in the heart, kidney, spleen, testis, and ovary [44]. It was found that *wt1a* was expressed in the kidney, ovary, and spleen of adult eels [55]. In our present study, the expression profile of *Lcwt1a* and *Lcwt1b* were examined in a panel of adult fish tissues via RT-qPCR, and the results showed that *Lcwt1a* mRNA was widely expressed in the kidney, spleen, ovary, and testis, while *Lcwt1b* was expressed mainly in the testis and then the ovary. Thus, here, we just focused on investigating the *Lcwt1b* gene in Asian seabass. Additionally, *Lcwt1b* showed a different expression pattern in adult tissue from those of other species, which may be due to the Asian seabass being a male precocious hermaphrodite fish; however, to address these issues, more extensive investigations on *Lcwt1b* gene will be carried out in our future work. In other words, *wtla* and *wt1b* are genes that are paralogous with each other, and their expression patterns showed some divergences but also some consistency in different teleosts, which suggests that they may both both be involved in the development of gonads and other organs in different fish species with various roles.

ISH showed that the signal of *Lczar1* was obvious in oogonia and primary-growth-stage oocytes, and was almost undetectable in vitellogenic-stage oocytes. Specifically, the signal was strongest in primary-growth-stage oocytes. It is indicated that *Lczar1* may play a crucial role in the early stage of oogenesis. The expression profile of *Lczar1* is similar to that in the early oocytes of zebrafish [18]. In the frog ovary, *zar1* is also detected in early oocytes [12]. In mice, *zar1* mRNA is detected in the early primary follicle through the antral follicle stages, but not in primordial follicles [11]. In *Xenopus laevis*, the expression level of *zar1* reaches the highest level in oocytes of stage I to III, and then decreases in stages IV to VI [17]. In rainbow trout, *zar1* is mainly expressed in metaphase II oocytes [26]. At the same time, in a study of spotted scat, it was found that the expression of *zar1* gradually increases with the development of the ovary, reaching its highest expression in Phase IV ovaries [27]. In Japanese eel, the expression of *zar1* gradually increases after ovarian differentiation, and is relatively high in Phase III–V ovaries [28]. These studies show that *zar1* in different species is expressed in early oocytes, including monoecious fish such as Asian seabass, which reveals that zar1 plays a vital role in early oogenesis.

On the contrary, *Lcwt1b* was predominantly expressed in the testis. Particularly, *Lcwt1b* was expressed in the somatic cells and germ cells of the ovary and testis, respectively. In mice, it was found that the expression of *wt1b* was detected in the embryonic mesoderm (such as the genitourinary ridge, gonad, and mesonephros) [56]. In mouse gonads, *wt1* is expressed in follicular granulosa cells, ovarian epithelial cell layer and oviduct fimbria, and also in Sertoli cells (SCs) in testicular convoluted seminiferous tubules [57]. In humans, *wt1* is only expressed by SCs in the adult testicular seminiferous epithelium [58], but also by endothelial cells of micro vessels and mesenchymal cells in fetal and neonatal testicular stroma [59]. In the testis of adult Medaka, *wt1a* expression is detected in somatic cells probably in SCs, whereas *wt1b* mRNA transcripts are restricted to the somatic cells of the testis periphery, being either germ cell-supporting cells or SCs [44]. Recently, Chen et al. believed that granulosa cells in the ovary and SCs in the testis are differentiated from the somatic cells of the genital ridge under the guidance of *wt1* [9]. As a female homozygous fish, *Lcwt1b* is expressed in somatic cells of testis and ovary, indicating that it may play an imperative role in the whole life cycle of Asian seabass, but the mechanism behind this in the process of sex reversal of Asian seabass is unclear. In addition, *Lcwt1b* was also expressed in oogonia and primary-growth-stage oocytes in the ovary, and in spermatocytes and spermatids in the testis, but the expression of *Lcwt1b* gradually decreased with the development of gametogenesis. Therefore, *Lczar1* and *Lcwt1b* may work in the early stage of oogenesis and spermatogenesis.

## 5. Conclusions

In summary, *Lczar1* and *Lcwt1b* were predominantly expressed in gonads and speculated to function in the gametogenesis of the androgynous fish Asian seabass. Our findings will lay a foundation for studying the molecular mechanisms of germ cell development, gametogenesis, and sex reversal in hermaphroditic fish.

## Figures and Tables

**Figure 1 animals-14-00508-f001:**
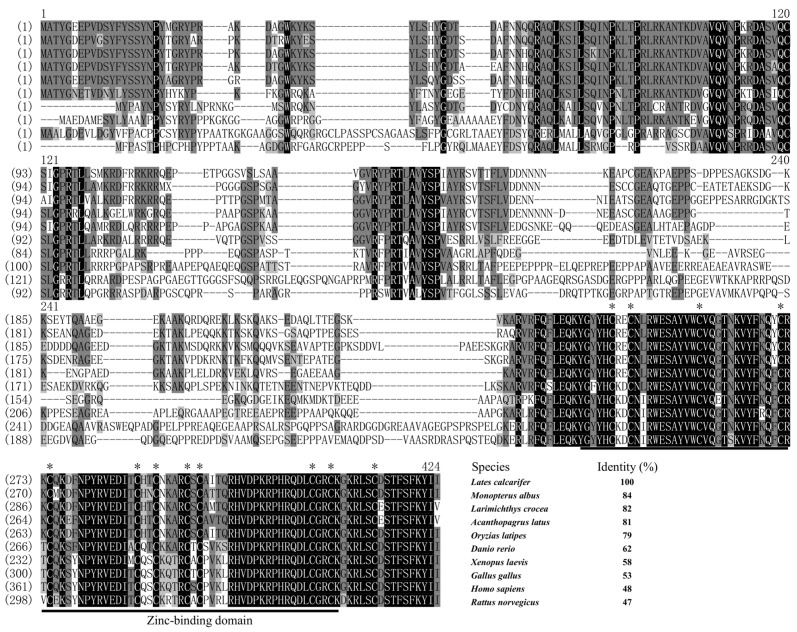
Multiple-sequence alignment analysis of Zar1 proteins in vertebrates. Black shadow: 100% identity; Grey shadow: 75% ≤ identity < 100%. Black underline: Zinc-binding domain. The positions shown by asterisks (*) are conservative C12 structures. At the end of the comparison, the species name and the identity percentage of Zar1 and its homologues were displayed. GenBank accession numbers for Zar1 are as follows: *Monopterus albus*: XP_020459078.1; *Acanthopagrus latus*: XP_036969674.1; *Larimichthys crocea*: XP_010729299.3; *Oryzias latipes*: XP_004067611.1; *Danio rerio*: NP_919362.2; *Xenopus laevis*: NP_001083958.1; *Gallus gallus*: XP_040527566.1; *Homo sapiens*: NP_783318.1: *Rattus norvegicus*: NP_852050.1.

**Figure 2 animals-14-00508-f002:**
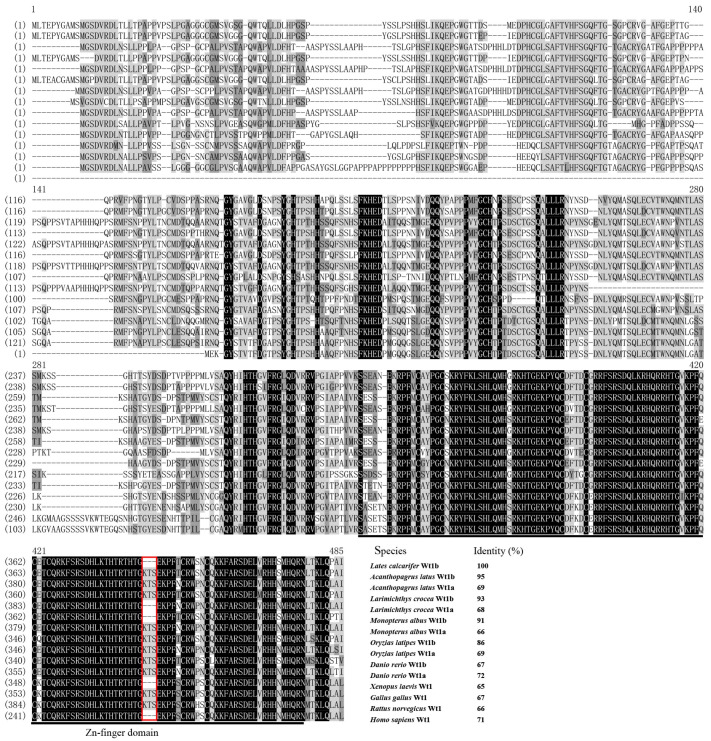
Multiple sequence alignment analysis of Wt1 proteins in vertebrates. Black shadow: 100% identity; grey shadow: 75% ≤ identity < 100%; light-grey shadow: 50% ≤ identity < 75%. Black underline: Zn-finger. Red box: Wt1(-KTS) or Wt1(+KTS). At the end of the comparison, the species name and the identity percentage of Wt1 and its homologues were displayed. GenBank accession numbers for Wt1 are as follows: *Monopterus albus* Wt1a: XP_020460123.1; *Monopterus albus* Wt1b: XP_020471332.1; *Acanthopagrus latus* Wt1a: XP_036963987.1; *Acanthopagrus latus* Wt1b: XP_036951482.1; *Larimichthys crocea* Wt1a: XP_027128154.1; *Larimichthys crocea* Wt1b: XP_027137243.1; *Oryzias latipes* Wt1a: ABG36853.1; *Oryzias latipes* Wt1b: ABG36854.1; *Danio rerio* Wt1a: AAI62638.1; *Danio rerio* Wt1b: AAI24088.1; *Xenopus laevis* Wt1: NP_001079057.1; *Gallus gallus* Wt1: NP_990547.2; *Rattus norvegicus* Wt1: NP_113722.2; *Homo sapiens* Wt1: AAH32861.1.

**Figure 3 animals-14-00508-f003:**
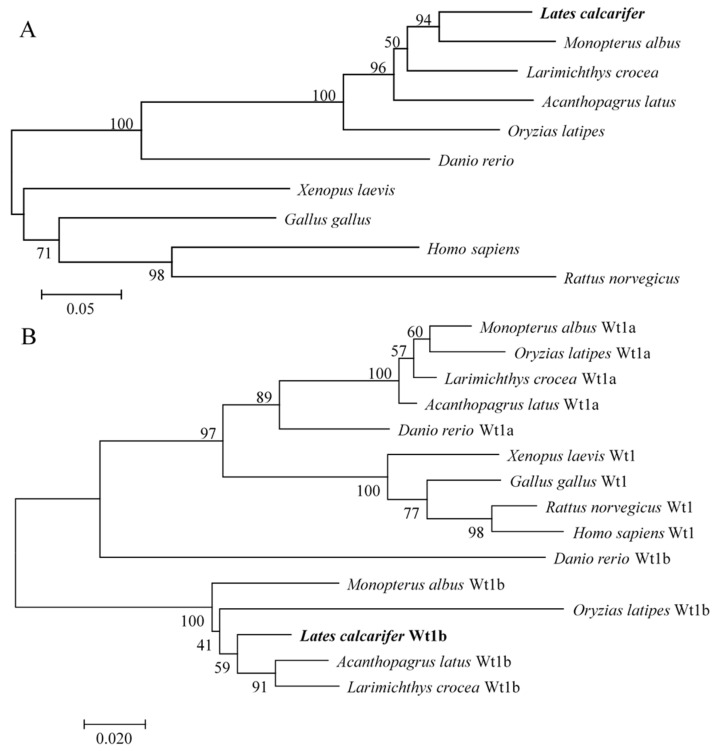
Phylogenetic trees of Zar1 (**A**) and Wt1 (**B**) proteins. Phylogenetic trees were deduced using MEGA 7.0 using the neighbor-joining method with 1000 bootstrap replicates. Numerals at the bifurcation points of the phylogenetic tree are bootstrap values. GenBank accession numbers for Zar1 and Wt1 are the same as those in Figure 1 and Figure 2.

**Figure 4 animals-14-00508-f004:**
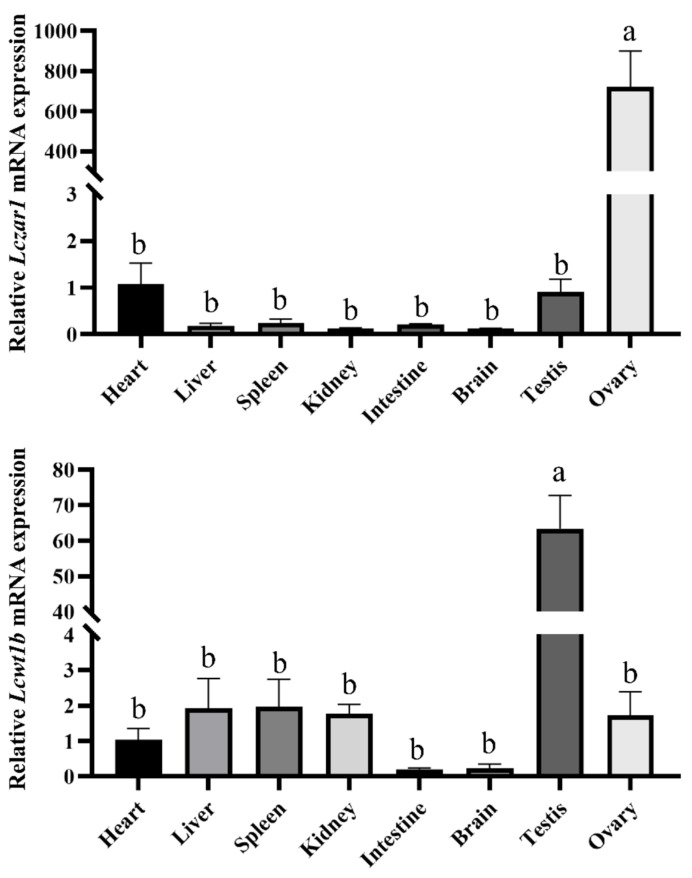
The mRNA expression profiles of *Lczar1* and *Lcwt1b* in adult Seabass tissues. RT-qPCR analyses were conducted, and *Lcβ-actin* was used as the internal reference gene. The relative mRNA expression levels of target genes were calculated using 2^−∆∆CT^, and the control was set as the heart tissue. Each column represents the means ± standard deviations (SD) from three biological replicates. Different lowercase letters indicated a significant difference at *p* < 0.05 via the method of one-way ANOVA and Tukey in SPSS 20.0.

**Figure 5 animals-14-00508-f005:**
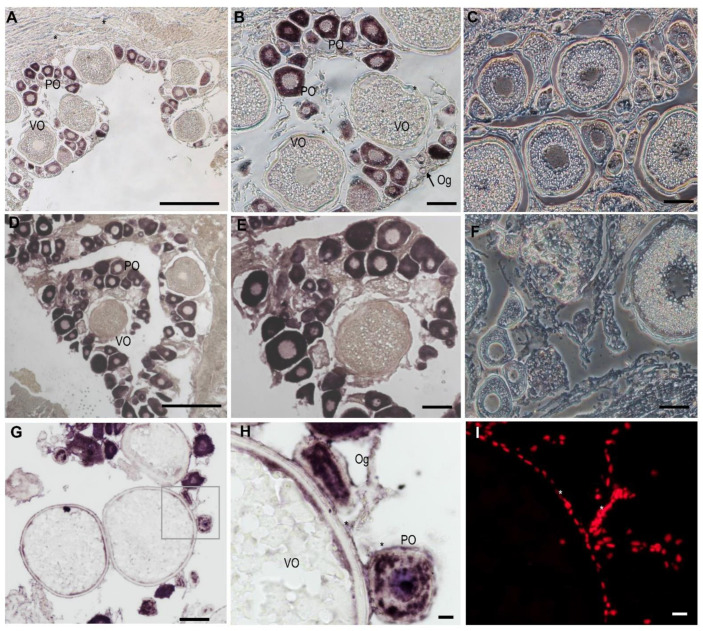
Cellular distribution of *Lczar1* and *Lcwt1b* mRNA in the adult ovary of the Asian seabass. The chemical ISH analysis were performed on the ovarian sections (signals in purple). (**A**,**B**) Antisense probe signal of *Lczar1*; (**C**) sense probe signal of *Lczar1*; (**D,E**,**G**,**H**) antisense probe signal of *Lczar1*; (**F**) sense probe signal of *Lcwt1b*; (**H**) antisense probe signal amplified from the frame of (**G**); (**I**) nuclei of ovarian cells counterstained with PI (in red). Og, oogonia; PO, primary-growth-stage oocytes; VO, vitellogenic-stage oocytes; asterisk (*), somatic cells. Scale bars: 50 μm.

**Figure 6 animals-14-00508-f006:**
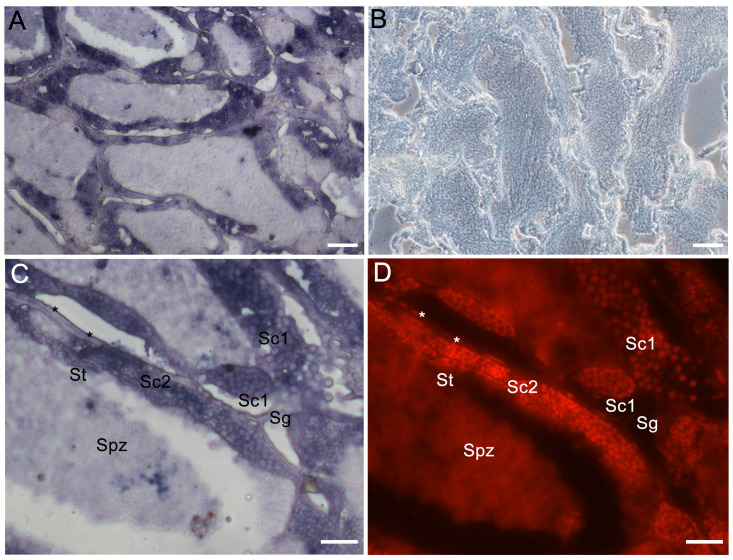
Cellular distribution of *Lcwt1b* mRNA in the adult testis of the Asian seabass. The chemical ISH analysis was performed on testicular sections (signals in purple). (**A**,**C**) Antisense probe signal of *Lcwt1b*; (**B**) sense probe signal of *Lcwt1b*; (**D**) nuclei of testicular cells counterstained with PI (in red). Sg, spermatogonia; Sc1, primary spermatocytes; Sc2, secondary spermatocytes; St, spermatids; Spz, spermatozoa; asterisk (*), somatic cells. Scale bars: 50 μm in (**A**,**B**); 20 μm in (**C**,**D**).

**Table 1 animals-14-00508-t001:** The sequences of primers used in this study.

Primer Name		Sequences (5′ to 3′)	Product Length (bp)	Tm (°C)	Purpose
*Lczar1-*CDS	Forward	CGGTTGGTTGAACAAAATGGC	1192	56	ISH, RT-PCR
	Reverse	GTGCGGTTCTCACCATCAAT			
*Lcwt1b-*CDS	Forward	ACTGTCTCAAACCGCCTTCA	1521	56	ISH, RT-PCR
	Reverse	GCAAGCACTAGTTGAGGTGC			
*Lczar1-*DL	Forward	CTACGATGGGAGAGTGCCTATG	247	60	RT-qPCR
	Reverse	AGCTGAAAGTGCTGTCGCAGG			
*Lcwt1b-*DL	Forward	ACATGAGGACACCCTGTCACC	289	60	RT-qPCR
	Reverse	GTACTGCGCACTGACGAGCAT			
*Lcβ-actin-*DL	Forward	CGGAATCCACGAGACCACCTAC	268	60	RT-qPCR
	Reverse	ACTCCTGCTTGCTGATCCACAT			

## Data Availability

The data that support the findings of this study are included in the manuscript or Appendix A.

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
