# Peer review of "Characterization of Two Gonadal Genes, zar1 and wt1b, in Hermaphroditic Fish Asian Seabass (Lates calcarifer)"

_animals, 2024, doi:10.3390/ani14030508_

Round 1
Reviewer 1 Report
Comments and Suggestions for Authors
The MS investigated the characterization of two gonadal genes zar1 and wt1b in hermaphroditic fish Asian seabass (Lates calcarifer) from gene clone, qRT-PCR, and in situ hybridization. These findings provide a basis for clarifying the mechanism of zar1 and wt1 in regulating germ cell development and sexual reverse of Asian seabass, which has a great significance. However, there are still some minor problems in the MS that need further revision as follows:
“Sertoli cells and granulosa cells” in L10 should not be placed in the functional section, it should be expressed as “zar1 and wt1 expressed in Sertoli cells and granulosa cells”.
All genes need to be given their full name when they first appear in the MS.
Gene only appear in one format, “Wt1b” or “wt1b”. And the whole MS needs to be unified.
If only wt1b exists in Asian seabass (Lates calcarifer), or if the MS only focuses on wt1b, it needs to be clarified and added to the discussion.
There are many format problems of gene in the MS, which need to be further unified.
Comments on the Quality of English Language
There are many format problems of gene in the MS, which need to be further unified.
Author Response
Question(Q) 1: “Sertoli cells and granulosa cells” in L10 should not be placed in the functional section, it should be expressed as “zar1 and wt1 expressed in Sertoli cells and granulosa cells”.
Response(R)1:Thanks for the reviewer’s good suggestion, we have corrected the original expression “Zygote arrest-1 (Zar1) plays a vital role in the process of oogenesis and Wilms’ tumor 1 (Wt1) not only involves in the process of oogenesis, but also exerts important functions in spermatogenesis, testicular development, gender differentiation, Sertoli cells and granulosa cells.” into “zygote arrest-1 (Zar1) and Wilms’ tumor 1 (Wt1) play an important role in oogeniesis, with the latter also involved in testicular development and gender differentiation.” following another reviewer’s suggestion.
Q2: All genes need to be given their full name when they first appear in the MS.
R2: We have given the full name of all genes and they are marked by red color.
Q3: Gene only appear in one format, “Wt1b” or “wt1b”. And the whole MS needs to be unified.
R3: We have unified all the gene names. We have unified the gene names into all lowercase in italic, while gene name appeared at the beginning of the sentences, the first letter of gene name was in capital. The protein name was presented in block with its first letter in capital.
Q4: If only wt1b exists in Asian seabass (Lates calcarifer), or if the MS only focuses on wt1b, it needs to be clarified and added to the discussion.
R4: Like other fish, both wt1a and wt1b exist in Asian seabass (Lates calcarifer). Here, we only focus on the wt1b because it is involved in gonadal development and the wt1a was found to be widely expressed adult fish tissues, including kidney, spleen, ovary and testis (data not shown). And we have adding the related descriptions in the discussion “In our present study, the expression profile of Lcwt1a and Lcwt1b were examined in a panel of adult fish tissues via RT-qPCR, the results showed that Lcwt1a mRNA was widely expressed in the kidney, spleen, ovary and testis, while Lcwt1b was expressed mainly in testis and then ovary. Thus, here, we just focused on investigating the Lcwt1b gene in Asian seabass.” in line 309-313.
Q5: There are many format problems of gene in the MS, which need to be further unified.
R5: Thanks for your suggestions, and we have unified the format of gene names in our revised MS.

Reviewer 2 Report
Comments and Suggestions for Authors
The authors describe the cloning and expression of two genes zar1 and wt1b from a protandrous hermaphrodite fish, Lates calcarifer. The work is of interest to the fish reproductive biology research community. However, feel the MS is very poorly written and therefore distracts the reader from the core findings of the research. For example, the convention of presenting gene names is not consistent nor follows the conventions consistently.
There is a tendency to repeat M&M in Results section and Results in the Discussion. These sections need to be carefully edited. From a technical point of view, I failed to understand why the study chose to clone only one of the isoforms, although both seem to have two isoforms in fish. In this connection any discussion with mammalian homologues doesn't make much sense.
Overall there are far too many inconsistencies in the current version. My recommendation is that it is subject to major revision. Attached below is marked PDF, with specific comments, that may be useful in revising the manuscript.

Needs significant improvement.
Author Response
Q1: However, feel the MS is very poorly written and therefore distracts the reader from the core findings of the research. For example, the convention of presenting gene names is not consistent nor follows the conventions consistently.
R1: We have unified all the gene names. We have unified the gene names into all lowercase in italic, while gene name appeared at the beginning of the sentences, the first letter of gene name was in capital. The protein name was presented in block with its first letter in capital.
Q2: There is a tendency to repeat M&M in Results section and Results in the Discussion. These sections need to be carefully edited. From a technical point of view, I failed to understand why the study chose to clone only one of the isoforms, although both seem to have two isoforms in fish. In this connection any discussion with mammalian homologues doesn't make much sense.
R2: We have rewritten the problems of repeat M&M in Results section and Results in the Discussion, which was marked by the red color.
R2: We have rewritten the related section to avoid problems of repeat M&M in Results section and Results in the Discussion, which were highlighted in the red color.
Like other fish, both wt1a and wt1b exist in Asian seabass (Lates calcarifer). Here, we only focus on the wt1b because it is involved in gonadal development and the wt1a was found to be widely expressed adult fish tissues, including kidney, spleen, ovary and testis (data not shown). And we have adding the related descriptions in the discussion “In our present study, the expression profile of Lcwt1a and Lcwt1b were examined in a panel of adult fish tissues via RT-qPCR, the results showed that Lcwt1a mRNA was widely expressed in the kidney, spleen, ovary and testis, while Lcwt1b was expressed mainly in testis and then ovary. Thus, here, we just focused on investigating the Lcwt1b gene in Asian seabass.” in line 309-313.
Q3: Overall there are far too many inconsistencies in the current version. My recommendation is that it is subject to major revision. Attached below is marked PDF, with specific comments, that may be useful in revising the manuscript.
R3:Thanks for the reviewer’s careful reading and the valuable suggestions. We have revised the manuscript point by point as required.

Reviewer 3 Report
Comments and Suggestions for Authors
Dear Authors,
In this manuscript titled “Characterization of two gonadal genes zar1 and wt1b in hermaphrodicit fish Asian seabass (Lates calcarifer), authors isolated two genes: zar1 and wt1b, and analyzed the expressions by qRT-PCR and in situ hybridization.
## Major Comments
1) Figure 7 is not included in this manuscript. The figure may show the expression patterns of Lczar1 and Lcwt1b genes in testis.
2) In this paper, authors characterized the two genes: zar1 and wt1b. However, reviewers could not find the proper reason why the author focused on these genes. What are authors interested in? How do these genes relate to that interest? Authors should clearly mention these points in introduction section.
3) Authors revealed the sequences of Lczar1 and Lcwt1b. However, the information is restricted, not a full-length sequence. Identification of these genes are important point in this paper. Therefore, it is essential to reveal the full-length sequences of these genes by 5’RACE PCR and 3’RACE PCR. In addition, authors should isolate the wt1a ortholog in Asian seabass (Lcwt1a).
4) Authors overemphasize the conclusion. For example, based on the only expression data, authors mentioned that “these results indicated that Lczar1 and Lcwt1b would play important roles in the gonadal development (at line 20 page 1)” and that “play a key role in the gametogenesis (at line 344 page 12)”. But, in order to indicate the roles of these genes, it is essential to perform the knockout or knockdown experiments. In this paper, authors indicate only expression patterns of Lczar1 and Lcwt1, not their roles. It may be difficult or impossible to perform the functional analysis in this species. Therefore, authors should weaken the wording of the conclusion. For example, “We speculated that…”, “It is possible that…”, or so on.
5) Line 262 page 11.
“the identity of zinc finger structures between 262 LcWt1b and M. albus Wt1b is 100%, which may be due to the fact that they are both her- 263 hermaphroditic fish.”
This sentence is also an unsubstantiated claim. This is a research paper. However, authors claim without evidence. In research papers, authors must clearly distinguish what can be indicated or shown with evidence from what is speculated. With this point in mind, authors should check again.
## Minor Comments
Figure 5: y-axis should indicate log2 scale. Data were presented as means ± standard error (SE). The number of biological replicates should be shown in the figure legend.
Line 229 page 9: What is CISH?
Line 252 page 10: Somatic cell types, such as Sertoli cells, Leydig cells, or so on, should be described.
Author Response
## Major Comments
Q1: Figure 7 is not included in this manuscript. The figure may show the expression patterns of Lczar1 and Lcwt1b genes in testis.
R1: It is true, the Figure 7 is not included in this manuscript and which shows the expression patterns of Lcwt1b genes in testis. We apologize for the technical error that happened in the pre-layout process. We have added this figure to the revised manuscript.
Q2: In this paper, authors characterized the two genes: zar1 and wt1b. However, reviewers could not find the proper reason why the author focused on these genes. What are authors interested in? How do these genes relate to that interest? Authors should clearly mention these points in introduction section.
R2: Thanks the reviewer’s comments. We have added the explanation for that we focused on the two genes in introduction section in line 101-105, which is “Based on the previous study, zar1 was found to be exclusively expressed in ovary while wt1b was predominantly involved in testicular development and sex differentiation, thus, investigations on zar1 and wt1b will facilitate demonstrating the molecular mechanism behind sex reversal in fish species.”
Q3: Authors revealed the sequences of Lczar1 and Lcwt1b. However, the information is restricted, not a full-length sequence. Identification of these genes are important point in this paper. Therefore, it is essential to reveal the full-length sequences of these genes by 5’RACE PCR and 3’RACE PCR. In addition, authors should isolate the wt1a ortholog in Asian seabass (Lcwt1a).
R3: Good suggestions. We have isolated the full-length sequences of the two genes which contained the ORFs and 5’-, 3’UTRs, respectively. Besides, we have examined the expression patterns of Lcwt1a, it was found to be widely expressed in the kidney, spleen, ovary and testis, while Lcwt1b was primarily expressed in testis and then ovary. Thus, we focused on studying the Lcwt1b gene in this study.
R4: Authors overemphasize the conclusion. For example, based on the only expression data, authors mentioned that “these results indicated that Lczar1 and Lcwt1b would play important roles in the gonadal development (at line 20 page 1)” and that “play a key role in the gametogenesis (at line 344 page 12)”. But, in order to indicate the roles of these genes, it is essential to perform the knockout or knockdown experiments. In this paper, authors indicate only expression patterns of Lczar1 and Lcwt1, not their roles. It may be difficult or impossible to perform the functional analysis in this species. Therefore, authors should weaken the wording of the conclusion. For example, “We speculated that…”, “It is possible that…”, or so on.
R4: Valuable suggestions. We have weaken the wording of the conclusion and changed the original statement “these results indicated that Lczar1 and Lcwt1b would play important roles in the gonadal development (at line 20 page 1)” into “These results indicated that Lczar1 and Lcwt1b would possibly play roles in the gonadal development.”, and changed the original statement “In summary, Lczar1 and Lcwt1b were predominantly expressed in gonads and play a key role in the gametogenesis of the androgynous fish (at line 344 page 12)” into “In summary, Lczar1 and Lcwt1b were predominantly expressed in gonads and speculated to function in the gametogenesis of the androgynous fish ” in line 353-354.
Q5: Line 262 page 11.
“the identity of zinc finger structures between 262 LcWt1b and M. albus Wt1b is 100%, which may be due to the fact that they are both her- 263 hermaphroditic fish.”
This sentence is also an unsubstantiated claim. This is a research paper. However, authors claim without evidence. In research papers, authors must clearly distinguish what can be indicated or shown with evidence from what is speculated. With this point in mind, authors should check again.
R5: Thanks for your suggestions, we have deleted this sentence.
## Minor Comments
Q6: Figure 5: y-axis should indicate log2 scale. Data were presented as means ± standard error (SE). The number of biological replicates should be shown in the figure legend.
R6: Good suggestions. We have added the number of biological replicates in the figure legend.
Q7: Line 229 page 9: What is CISH?
R7: CISH represents Chemical in situ Hybridization, we have revised the CISH into chemical ISH.
Q8: Line 252 page 10: Somatic cell types, such as Sertoli cells, Leydig cells, or so on, should be described.
R8: Very good suggestions. We have added the type of somatic cell types in line 242 in the revised manuscript.

Round 2
Reviewer 3 Report
Comments and Suggestions for Authors
In R3, authors mentioned that full-length sequences of the two genes which contained the ORF, 5' UTR and 3' UTR were isolated. However, the sequences were contained only a partial sequences of 5' UTR and 3' UTR, not full-length sequences. It is essential to reveal the full-length sequences of these genes by 5’RACE PCR and 3’RACE PCR.
The quality of Figure 7, which is the expression data in testes, is very low. it is impossible to identify the cell types, such as spermatogonia, spermatocyte, Sertoli cells.
Author Response
Question 1:In R3, authors mentioned that full-length sequences of the two genes which contained the ORF, 5' UTR and 3' UTR were isolated. However, the sequences were contained only a partial sequences of 5' UTR and 3' UTR, not full-length sequences. It is essential to reveal the full-length sequences of these genes by 5’RACE PCR and 3’RACE PCR.
Answer 1: Thanks the reviewer’s suggestion very much, we have added the word “partial” before the 5' UTR and 3' UTR and highlighted in line 170-178 to illustrate that we have obtained the partial cDNA fragments of the two genes. We will isolate the full-length sequences of these genes by 5’RACE PCR and 3’RACE PCR in our further study.
Question 2: The quality of Figure 7, which is the expression data in testes, is very low. it is impossible to identify the cell types, such as spermatogonia, spermatocyte, Sertoli cells.
Answer 2: We have provided the high quality of all Figures, thanks the reviewer’s good suggestion.